# New Polyene Macrolide Compounds from Mangrove-Derived Strain *Streptomyces hiroshimensis* GXIMD 06359: Isolation, Antifungal Activity, and Mechanism against *Talaromyces marneffei*

**DOI:** 10.3390/md22010038

**Published:** 2024-01-08

**Authors:** Zhou Wang, Jianglin Yin, Meng Bai, Jie Yang, Cuiping Jiang, Xiangxi Yi, Yonghong Liu, Chenghai Gao

**Affiliations:** 1Institute of Marine Drugs, Guangxi University of Chinese Medicine, Nanning 530200, China; wangzhou2021@stu.gxtcmu.edu.cn (Z.W.); yinjianglin@126.com (J.Y.); xxbai2014@163.com (M.B.); jieyang202312@163.com (J.Y.); ping990120@foxmail.com (C.J.); 2Guangxi Key Laboratory of Marine Drugs, Guangxi University of Chinese Medicine, Nanning 530200, China; yixiangxi2017@163.com; 3Guangxi Scientific Research Center of Traditional Chinese Medicine, Nanning 530200, China

**Keywords:** actinomycetes, polyene macrolide, *Streptomyces hiroshimensis*, antifungal activity, antifungal mechanism, *Talaromyces marneffei*

## Abstract

Mangrove-derived actinomycetes represent a rich source of novel bioactive natural products in drug discovery. In this study, four new polyene macrolide antibiotics antifungalmycin B-E (**1**–**4**), along with seven known analogs (**5**–**11**), were isolated from the fermentation broth of the mangrove strain *Streptomyces hiroshimensis* GXIMD 06359. All compounds from this strain were purified using semi-preparative HPLC and Sephadex LH-20 gel filtration while following an antifungal activity-guided fractionation. Their structures were elucidated through spectroscopic techniques including UV, HR-ESI-MS, and NMR. These compounds exhibited broad-spectrum antifungal activity against *Talaromyces marneffei* with minimum inhibitory concentration (MIC) values being in the range of 2–128 μg/mL except compound **2**. This is the first report of polyene derivatives produced by *S. hiroshimensis* as bioactive compounds against *T*. *marneffei*. In vitro studies showed that compound **1** exerted a significantly stronger antifungal activity against *T*. *marneffei* than other new compounds, and the antifungal mechanism of compound **1** may be related to the disrupted cell membrane, which causes mitochondrial dysfunction, resulting in leakage of intracellular biological components, and subsequently, cell death. Taken together, this study provides a basis for compound **1** preventing and controlling talaromycosis.

## 1. Introduction

*Talaromyces* (*Penicillium*) *marneffei* causes a life-threatening mycosis—Talaromycosis, an endemic invasive mycosis found primarily in tropical and subtropical Southeast Asia [1,2,3]. In recent years, some drugs have been reported to encounter antifungal resistance for *T*. *marneffei*, especially fluconazole [4,5,6,7,8,9]. Moreover, the disease is being recognized with an increasing frequency well beyond the original endemic areas [10,11,12,13,14,15,16]. Therefore, the discovery of new antifungal drugs is necessary to cover the shifting challenges of this and other fungal infections.

Mangroves are ecologically significant plants in marine habitats that inhabit the coastlines of many countries [17]. Being a highly productive and diverse ecosystem, mangroves are rich in numerous classes of actinomycete that are of great importance in the field of antibiotics [18]. Furthermore, about 200 compounds, such as salinosporamide A (to be processed for clinical trials for cancer treatment), xiamycins, rifamycins, and antimycin A were discovered from mangrove actinobacteria, which have become an important source of novel bioactive compounds [19,20,21,22,23].

In our preliminary study, *Streptomyces hiroshimensis* GXIMD 06359 was isolated from mangrove in the west coast of Hainan [24], which was identified as the potential strain to produce antimicrobial active metabolites against *T*. *marneffei* in Am2ab medium. The scaled-up fermentation and extensive chromatographic separation of the EtOAc extract resulted in the isolation of four new metabolites, namely antifungalmycins B-E (**1**–**4**), together with seven known compounds (**5**–**11**). Herein, we report the isolation and structural determination of these compounds (Figure 1) along with the antifungal activities. Then the mechanism of compound **1** inhibiting *T*. *marneffei* was studied, with the aim of finding potential new drugs for the precision treatment of talaromycosis.

## 2. Results

### 2.1. Structural Elucidation

Compound **1** was obtained as a light-yellow amorphous powder. Its HR-ESI-MS spectrum showed a characteristic [M − H]^−^ ion peak at *m/z* 675.3578 (calcd for 675.3592), which is consistent with the molecular formula of C_33_H_56_O_14_, indicating 6 degrees of unsaturation. The ^1^H NMR spectrum (Appendix A) shows resonances for three methyl protons *δ*_H_ 1.14 (3H, d, *J* = 6.2 Hz, H_3_-28), 1.06 (3H, s, H_3_-29), and 0.84 (3H, t, *J* = 6.2 Hz, H_3_-6′); seven methylene protons *δ*_H_ 1.71 (1H, m, H-10a), 1.25 (1H, m, H-10b), 1.62 (2H, m, H_2_-12), *δ*_H_ 1.54 (2H, m, H_2_-6), 1.52 (2H, m, H_2_-4), 1.47 (2H, m, H_2_-8), 1.44 (1H, m, H-3′a), 1.25 (1H, m, H-3′b), 1.29 (1H, m, H-2′a), and 1.24 (1H, m, H-2′b); thirteen methine protons *δ*_H_ 4.71 (1H, m, H-27), 4.24 (1H, dd, *J* = 5.7, 2.3 Hz, H-25), 4.02 (1H, m, H-3), 3.97 (1H, m, H-9), 3.90 (1H, m, H-5), 3.90 (1H, m, H-7), 3.90 (1H, m, H-11), 3.68 (1H, m, H-1′), 3.65 (1H, m, H-15), 3.51 (1H, dd, *J* = 7.2, 2.6 Hz, H-26), 3.34 (1H, m, H-14), 3.29 (1H, m, H-13), and 2.46 (1H, dd, *J* = 8.6, 6.8 Hz, H-2); and eight olefinic protons *δ*_H_ 6.28 (1H, m, H-21), 6.28 (1H, m, H-22), 6.28 (1H, m, H-23), 6.26 (1H, m, H-20), 6.24 (1H, m, H-19), 6.15 (1H, dd, *J* = 15.1, 9.4 Hz, H-18), 5.72 (1H, d, *J* = 15.0 Hz, H-17), and 5.67 (1H, dd, *J* = 14.6, 5.8 Hz, H-24). The ^13^C NMR and DEPT 135 spectra (Appendix A) displayed thirty-three signals for carbon, including one carbonyl carbon *δ*_C_ 171.6 (C-1); one quaternary carbon *δ*_C_ 82.0 (C-16); eight olefinic carbons *δ*_C_ 138.9 (C-17), 134.1 (C-24), 133.5 (C-22), 133.2 (C-20), 131.5 (C-19), 131.2 (C-21), 130.4 (C-23), and 126.3 (C-18); thirteen methine carbons *δ*_C_ 82.7 (C-15), 82.3 (C-14), 75.7 (C-26), 75.2 (C-13), 72.0 (C-25), 70.3 (C-27), 69.6 (C-3), 69.2 (C-9), 69.2 (C-1′), 68.5 (C-7), 67.8 (C-5), 67.8 (C-11), and 58.3 (C-2); seven methylene carbons *δ*_C_ 44.5 (C-8), 44.3 (C-6), 42.2 (C-10), 41.6 (C-12), 40.9 (C-4), 36.7 (C-2′), and 18.2 (C-3′); and three methyl carbons *δ*_C_ 22.9 (C-29), 17.4 (C-28), and 14.0 (C-1′).

Careful analysis of the ^1^H and ^13^C NMR data (Table 1) of **1** showed they were very similar to antifungalmycin [25,26], which possesses the same lactone ring. The main difference was that the signals for two methylene groups [(*δ*_H_ 1.24, *δ*_C_ 31.3, CH_2_-4′) and (*δ*_H_ 1.25, *δ*_C_ 22.1, CH_2_-5′)] at the side chain in the NMR spectra of antifungalmycin were absent in compound **1**. Those were supported by the ^1^H-^1^H COSY and HMBC correlations (Figure 2). In the ^1^H-^1^H COSY spectrum of compound **1**, correlations were observed for H-2/H-3/H-1′/H-2′/H-3′/H-4′, H-10/H-11/H-12/H-13/H-14/H-15, and H-24/H-25/H-26/H-27. The HMBC spectrum shows correlations of H-2 to C-1/C-4/C-1′/C-2′, H-14 to C-12/C-13/C-15, H-15 to C-16/C-17/C-29, H-25 to C-23/C-24/C-25, H-26 to C27/C-28, and H-27 to C-1. Thus, the planar structure of **1** was determined. In the NOESY spectrum, the correlations of H-3/H-1′, H-13/H-15, H-25/H-26 (Figure 3) were not enough to elucidate the compound **1** relative configuration. Consequently, compound **1**’s relative configuration has not been elucidated, and is named antifungalmycin B.

Compound **2**, isolated as a light-yellow amorphous powder. The molecular formula was deduced to be C_33_H_56_O_11_ by the HR-ESI-MS peak at *m/z* 627.3743 [M−H]^−^ (calcd for 627.3744), indicating 6 degrees of unsaturation. The ^1^H NMR spectrum (Appendix A) shows resonances for three methyl protons *δ*_H_ 1.19 (3H, d, *J* = 6.3 Hz, H_3_-26), 0.84 (3H, m, H_3_-27), and 0.84 (3H, t, *J* = 6.9 Hz, H_3_-6′); eight methylene protons δ_H_ 1.92 (1H, m, H-12a), 1.52 (1H, m, H-12b), 1.50 (2H, m, H_2_-8), 1.42 (1H, m, H-3′a), 1.24 (1H, m, H-3′b), 1.38 (2H, m, H_2_-4), 1.40 (2H, m, H_2_-6), 1.70 (1H, m, H-10a), 1.25 (1H, m, H-10b), 1.34 (1H, m, H-2′a), 1.23 (1H, m, H-2′b), 1.24 (2H, m, H_2_-4′), and 1.24 (2H, m, H_2_-5′); twelve methine protons *δ*_H_ 4.62 (1H, m, H-25), 4.05 (1H, m, H-3), 3.93 (1H, m, H-5), 3.93 (1H, m, H-7), 3.93 (1H, m, H-9), 3.93 (1H, m, H-11), 3.82 (1H, m, H-24), 3.72 (1H, m, H-1′), 3.63 (1H, m, H-15), 3.21 (1H, m, H-13), 2.87 (1H, m, H-14), and 2.38 (1H, t, *J* = 8.0 Hz, H-2); and eight olefinic protons *δ*_H_ 6.28 (1H, m, H-19), 6.26 (1H, m, H-20), 6.23 (1H, m, H-21), 6.20 (1H, m, H-17), 6.20 (1H, m, H-22), 6.18 (1H, m, H-18), 5.89 (1H, dd, *J* = 14.0, 3.1 Hz, H-16), and 5.79 (1H, dd, *J* = 14.7, 6.8 Hz, H-23). The ^13^C NMR and DEPT 135 spectra (Appendix A) displayed thirty-three signals for carbon, including one carbonyl carbon *δ*_C_ 171.2 (C-1); eight olefinic carbons *δ*_C_ 135.9 (C-23), 133.8 (C-22), 133.3 (C-17), 131.4 (C-19), 131.1 (C-21), 130.7 (C-20), 130.4 (C-18), and 129.5 (C-16); twelve methine carbons *δ*_C_ 80.5 (C-13), 76.1 (C-14), 73.3 (C-26), 73.0 (C-13), 72.0 (C-25), 70.3 (C-27), 70.2 (C-3), 69.8 (C-9), 69.7 (C-1′), 69.6 (C-7), 66.7 (C-5), and 58.5 (C-2); seven methylene carbons *δ*_C_ 44.0 (C-6), 43.7 (C-8), 43.3 (C-10), 40.9 (C-4), 38.5 (C-12), 33.9 (C-2′), 24.4 (C-3′), 31.3 (C-4′), and 22.1 (C-5′); and three methyl carbons *δ*_C_ 17.8 (C-26), 15.6 (C-27), and 14.0 (C-1′).

The ^1^H and ^13^C NMR data (Table 1) of **2** suggested that it was similar to antifungalmycin [25,26]. Combined with analyzing the ^1^H–^1^H COSY and HMBC spectra, it was revealed that compound **2** was a 26-membered macrocyclic lactone ring. Compared with antifungalmycin, the signals for one-quarter carbon (*δ*_C_ 82.0) and one methylene were absent in the lactone ring in compound **2**. In the COSY spectrum, correlations were observed for H-2/H-3/H-4/H-1′/H-2′/H-3′/H-4′/H-5′/H-6′, H-10/H-11/H-12/H-13/H-14/H-15/H-16, and H-23/H-24/H-25/H-26 (Figure 2). The HMBC spectrum shows correlations of H-13 to C-14/C-27, H-15 to C-14/C-16/C-27, H-23 to C-22/C-23/C-24, H-24 to C-22/C-25, and H-25 to C-1. Thus, the planner structure of **2** was confirmed. In the NOESY spectrum, the correlations of H-3/H-1′ and H-13/H-15 (Figure 3) were not enough to elucidate its relative configuration. Compound **2’s** relative configuration has not been elucidated, and is named antifungalmycin C.

Compound **3** was obtained as a light-yellow amorphous powder. Its HR-ESI-MS data ([M + H]^+^, 705.4088; calcd for C_35_H_61_O_14_, 705.4061) possessed the molecular formula of C_35_H_60_O_14_ (6 degrees of unsaturation). The ^1^H NMR spectrum (Appendix A) shows resonances for three methyl protons *δ*_H_ 1.21 (3H, d, *J* = 6.2 Hz, H_3_-28), 1.20 (3H, s, H_3_-29), and 0.85 (3H, t, *J* = 6.9 Hz, H_3_-6′); eight methylene protons *δ*_H_ 3.33 (2H, m, H_2_-12), 1.76 (1H, m, H-10a),1.45 (1H, m, H-10b), 1.46 (2H, m, H_2_-8), 1.45 (2H, m, H_2_-6), 1.45 (2H, m, H_2_-4), 1.45 (1H, m, H-2′a), 1.25 (1H, m, H-2′b), 1.45 (1H, m, H-3′a), 1.25 (1H, m, H-3′b), 1.25 (2H, m, H_2_-5′), and 1.24 (2H, m, H_2_-4′); twelve methine protons *δ*_H_ 4.57 (1H, dd, *J* = 5.1, 2.7 Hz, H-27), 4.05 (1H, m, H-3), 3.98 (1H, m, H-7), 3.93 (1H, m, H-5), 3.93 (1H, m, H-9), 3.93 (1H, m, H-11), 3.87 (1H, m, H-26), 3.80 (1H, m, H-14), 3.71 (1H, dd, *J* = 5.1, 2.7 Hz, H-15), 3.68 (1H, m, H-1′), 3.46 (1H, m, H-13), and 2.48 (1H, m, H-2); and eight olefinic protons *δ*_H_ 6.36 (1H, m, H-24), 6.31 (1H, m, H-19), 6.31 (1H, m, H-21), 6.26 (1H, m, H-20), 6.26 (1H, m, H-22), 6.23 (1H, m, H-23), 5.99 (1H, dd, *J* = 15.1, 5.1 Hz, H-23), and 5.72 (1H, dd, *J* = 14.5, 4.3 Hz, H-18). The ^13^C NMR (Appendix A) displayed thirty-five signals for carbon, including one carbonyl carbon *δ*_C_ 171.3 (C-1); one quaternary carbon *δ*_C_ 80.0 (C-16); eight olefinic carbons *δ*_C_ 129.1 (C-18), 131.1 (C-19), 130.4 (C-20), 131.5 (C-21), 131.2 (C-22), 133.1 (C-23), 133.9 (C-24), and 136.1 (C-25); twelve methine carbons *δ*_C_ 84.6 (C-14), 83.4 (C-17), 78.6 (C-15), 72.5 (C-27), 72.4 (C-26), 70.0 (C-3), 69.7 (C-1′), 69.0 (C-7), 68.2 (C-5), 67.0 (C-9), 66.9 (C-11), and 58.2 (C-2); eight methylene carbons *δ*_C_ 46.3 (C-8), 45.4 (C-6), 43.6 (C-10) 43.3 (C-4), 63.1 (C-12), 34.3 (C-2′), 24.6 (C-3′), 31.3 (C-4′), and 22.1 (C-5′); and three methyl carbons δ_C_ 17.9 (C-28), 19.6 (C-29), and 14.0 (C-1′).

The NMR data (Table 1) of **3** suggested that it was similar to antifungalmycin [25,26]. Detailed analysis of the ^1^H–^1^H COSY and HMBC spectra revealed that the main differences between them were the positions of conjugated double bonds. The conjugated double bonds of **3** were in C-18 to C-25, but in antifungalmycin were in C-17 to C-24. In the COSY spectrum, correlations were observed for H-2/H-3/H-4/H-1′/H-2′/H-3′/H-4′/H-5′/H-6′, H-13/H-14/H-15, H-17/H-18, and H-25/H-26/H-27/H-28 (Figure 2). The HMBC spectrum shows correlations of H-2 to C-1/C-3/C-1′/C-2′, H-14 to C-13/C-14, H-15 to C-16/C-17, H-17 to C-19, H-18 to C-17/C-19, H-25 to C-24/C-26/C-27, and H-29 to C-17 (Figure 2). Thus, the planar structure of **3** was determined. In the NOESY spectrum, the correlations of H-3/H-1′, H-13/H-14/H-15 (Figure 3) were not enough to elucidate the compound **3** relative configuration. Consequently, compound **3’s** relative configuration has not been elucidated, and is named antifungalmycin D.

Compound **4** was obtained as a light-yellow amorphous powder. The molecular formula was determined as C_35_H_60_O_13_ (five degrees of unsaturation) via HR-ESI-MS (*m/z* 687.3975, [M − H]^−^, (C_35_H_59_O_13_ calcd. 687.3956)). The ^1^H NMR spectrum (Appendix A) shows resonances for three methyl protons *δ*_H_ 1.14 (3H, d, *J* = 6.3 Hz, H_3_-28), 1.10 (3H,s, H_3_-29), and 0.84 (3H, t, *J* = 7.1 Hz, H_3_-6′); seven methylene protons *δ*_H_ 1.76 (1H, m, H-10a), 1.53 (1H, m, H-10b), 1.74 (1H, m, H-12a),1.50 (1H, m, H-12b), 1.50 (2H, m, H_2_-4), 1.50 (2H, m, H_2_-6), 1.50 (2H, m, H_2_-8), 1.42 (1H, m, H-3′a), 1.24 (1H, m, H-3′b), 1.33 (1H, m, H-2′a), 1.24 (1H, m, H-2′b), 1.25 (2H, m, H_2_-5′), and 1.24 (2H, m, H_2_-4′); thirteen methine protons *δ*_H_ 4.77 (1H, m, H-27), 4.23 (1H, dd, *J* = 5.7, 2.3 Hz, H-25), 4.03 (1H, m, H-3), 3.97 (1H, m, H-9), 3.92 (1H, m, H-5), 3.92 (1H, m, H-7), 3.90 (1H, m, H-15), 3.87 (1H, m, H-11), 3.65 (1H, m, H-13), 3.65 (1H, m, H-1′), 3.50,(1H, m, H-26), 2.46 (1H, dd, *J* = 8.8, 6.6 Hz, H-2), and 2.16 (1H, m, H-14); and eight olefinic protons *δ*_H_ 6.32 (1H, m, H-22), 6.25 (1H, m, H-23), 6.22 (1H, m, H-19), 6.21 (1H, m, H-20), 6.21 (1H, m, H-21), 6.15 (1H, dd, *J* = 14.5, 6.2 Hz, H-18), 5.68 (1H, dd, *J* = 14.6, 6.1 Hz, H-24), and 5.65 (1H, d, *J* = 15.0 Hz, H-17). The ^13^C NMR and DEPT 135 spectra (Appendix A) displayed thirty-five signals for carbon, including one carbonyl carbon *δ*_C_ 171.7 (C-1); one quaternary carbon *δ*_C_ 84.5 (C-16); eight olefinic carbons *δ*_C_ 137.8 (C-17), 134.4 (C-24), 134.0 (C-22), 133.5 (C-21), 131.1 (C-20), 130.8 (C-19), 130.3 (C-23), and 127.4 (C-18); eleven methine carbons *δ*_C_ 76.0 (C-15), 75.5 (C-26), 72.1 (C-25), 71.2 (C-13), 70.2 (C-27), 69.4 (C-3), 69.4 (C-5), 69.2 (C-1′), 68.2 (C-7), 67.5 (C-11), and 58.3 (C-2); seven methylene carbons *δ*_C_ 44.6 (C-8), 44.4 (C-12), 44.1 (C-6), 41.9 (C-14), 41.4 (C-10), 40.9 (C-4), 34.5 (C-2′), 24.5 (C-3′), 31.3 (C-4′), and 22.1 (C-5′); and three methyl carbons *δ*_C_ 22.2 (C-29), 17.5 (C-28), and 14.0 (C-1′).

The ^1^H and ^13^C NMR data (Table 1) of compound **4** were very similar to antifungalmycin [25,26]. Detailed analysis of the ^1^H–^1^H COSY and HMBC spectra revealed that compound **4** has no hydroxyl substitution in the C-14 (*δ*_H_ 2.16/*δ*_C_ 41.9). In the COSY spectrum, correlations were observed for H-2/H-3/H-1′/H-2′/H-3′/H-4′/H-5′/H-6′, H-10/H-11/H-12/H-13/H-14/H-15, and H-24/H-25/H-26/H-27/H-28 (Figure 2). The HMBC spectrum shows correlations of H-2 to C-1/C-3/C-4/C-1′/C-2′, H-15 to C-13/C-16/C-17, H-25 to C-22/C-23/C-24, and H-27 to C-1/C-25/C-26/C-28 (Figure 2). On the basis of the evidence, the planar structure of **4** was confirmed. In the NOESY spectrum, the correlations of H-3/H-1′, H-13/H-15, and H-25/H-26 (Figure 3) were not enough to elucidate the compound **4**’s relative configuration. Consequently, compound **4**’s relative configuration has not been elucidated, and is named antifungalmycin E.

The known compounds **5** and **6** had the same molecular formula of C_35_H_59_O_14_ as determined by HR-ESI-MS. The ^1^H and ^13^C NMR data of compounds **5** and **6** revealed that they were similar to the previously reported antifungalmycin [25,26]. The differences between **5** and **6** were in the positions C-1 (*δ*_C_ 171.8 vs. 171.1), C-2 (*δ*_C_ 57.5 vs. 58.2), C-3 (*δ*_C_ 69.7 vs. 69.2), C-1′ (*δ*_C_ 69.5 vs. 68.2), 2-H (*δ*_H_ 2.46 vs. 2.32), and 1′-OH (*δ*_H_ 4.81 vs. 4.45). The above difference in chemical shift may be caused by the different stereoscopic configuration of C-3 and C-1′. Thus, compound **5** was named antifungalmycin a_1_ and compound **6** was named antifungalmycin a_2_.

The molecular formulas of compounds **7** and **8** were determined to be C_35_H_58_O_12_ by HR-ESI-MS. They were identified as fungichromin [27], by comparing NMR and HR-ESI-MS data with reported values. The differences between **7** and **8** were in the positions C-1 (*δ*_C_ 171.1 vs. 170.5), C-2 (*δ*_C_ 58.7 vs. 57.8), C-3 (*δ*_C_ 70.0 vs. 69.4), C-1′ (*δ*_C_ 69.6 vs. 68.2), 2-H (*δ*_H_ 2.46 vs. 2.31), and 1′-OH (*δ*_H_ 4.81 vs. 4.38). The difference in chemical shift may be caused by the different stereoscopic configuration of C-3 and C-1′. Thus, compound **7** was named fungichromin a_1_ and compound **8** was named fungichromin a_2_.

The known compounds **9** and **10** were identified as filipin III by comparing their NMR and HR-ESI-MS data with those previously reported [28]. The differences between **9** and **10** were in the positions C-1 (*δ*_C_ 171.1 vs. 170.5), C-2 (*δ*_C_ 58.7 vs. 57.8), C-3 (*δ*_C_ 70.0 vs. 69.7), C-1′ (*δ*_C_ 69.6 vs. 67.9), 2-H (*δ*_H_ 2.46 vs. 2.29), and 1′-OH (*δ*_H_ 4.89 vs. 4.42). The above difference in chemical shift may be caused by the different stereoscopic configuration of C-3 and C-1′. Thus, compound **9** was named filipin III a_1_ and compound **10** was named filipin III a_2_.

The known compound **11** was obtained as a light-yellow amorphous powder. It was identified as filipin I [29], by comparing ^1^H-NMR, ^13^C-NMR, and HR-ESI-MS data with that reported.

### 2.2. Antifungal Activity of the Compounds

The antifungal potency of the compounds against *T*. *marneffei* was evaluated by MIC and MFC values; the results are shown in Table 2. In our experiment, most of the compounds exhibited antifungal activity except compound **2**, which has an MIC value of more than 128 μg/mL. In addition, compound **9** exhibited the best antifungal activity against *T*. *marneffei*, with MIC and MFC values of 2 and 4 μg/mL, respectively. Moreover, compound **1** showed the lowest MIC and MBC values, which represented the best antifungal activity of all the new compounds. So, we further explore the antifungal mechanism of compound **1** in the next experiment.

### 2.3. Compound ***1*** Inhibited the Growth of T. marneffei

To further analyze the growth-inhibiting characteristics of compound **1**, the time course of *T*. *marneffei* growth in the presence of **1** at different concentrations was plotted. *T*. *marneffei* exhibited rapid growth in the control group, and the logarithmic growth stage was achieved within 40 h of incubation, then entered a stabilization phase after 60 h of incubation (Figure 4A). However, the growth of *T*. *marneffei* after treatment with **1** at 1/2 MIC and 1 MIC showed a substantially lower growth rate than that of the control. Moreover, before 60 h of incubation, no further growth of *T. marneffei* was observed in 1/2 MIC and 1 MIC compound **1**-treated groups. But after 60 h of incubation, *T*. *marneffei* with **1** continued to grow, though at a lower rate than control, and the fungi growth rate of group compound **1** at 1 MIC was lower than group compound **1** at 1/2 MIC. Overall, these results confirmed that **1** had an inhibitory effect on *T*. *marneffei* growth, and was shown to be concentration-dependent and time-limited.

### 2.4. Compound ***1*** Disrupted the Cell Membrane of T. marneffei

The membrane integrity of the fungi was investigated to verify the ability of compound **1** to damage the fungal cell membrane. The electric conductivity of the cell suspensions implied the permeability of the cell membrane. Electrolytes are charged molecules such as sodium chloride and potassium chloride, and they are essential for fungal metabolism and growth [30]. Thus, their leakage can lead to fungal inhibition or death. Compared with the control, compound **1** resulted in a significant increase in conductivity. The conductivity of *T*. *marneffei* increased significantly from 1.80 for the control to 2.99 and 3.69 in the presence of compound **1** at the levels of 1/2 MIC and 1 MIC after 15 h, respectively (Figure 4B). Moreover, after 9 h exposure to the 1/2 MIC of compound **1**, the extracellular conductivity entered a steady stage. However, for the group of 1 MIC, the extracellular conductivity continued to increase. This indicates that compound **1** has a destructive effect on the cell membrane of *T*. *marneffei*, and shows a concentration dependence consistent with the growth-inhibiting results.

In order to further determine the degree of cell membrane damage by compound **1**, in the current work, nucleic acids and proteins released from the cytoplasm were monitored by the detection of absorbance at 260 nm and 280 nm, respectively. As the previous work reported, nucleic acid and protein play important roles in bacterial metabolism as they dominate the genetic information and cellular structure [31]. Leakage of cellular materials was analyzed by detecting 260 nm and 280 nm absorbing materials. Therefore, the absorbance of the material and proteins at 260 nm and 280 nm wavelengths can be used as an indicator of damage to the cell wall and membrane, which causes leakage of the cellular materials into the surroundings [32]. As shown in Figure 4C,D, both cell constituents were released rapidly from *T*. *marneffei* into cell suspensions and their amounts increased multi-fold after treatment with compound **1**. In addition, there was a progressive release of proteins and nucleic acids from *T*. *marneffei* after exposure to compound **1** for 4 h, followed by a steady state. Moreover, the leakage of nucleic acids and proteins in the group treated with 1 MIC compound 1 was larger than the control and 1/2 MIC compound **1** group. Compound **1** dose-dependently destroyed the cell membrane of *T. marneffei*, which was consistent with the previous results. Similar results have also been reported for the crude methanolic extract of *Myrtus communis* roots and leaves when tested against *Candida glabrata*, showing increased absorbance at a wavelength of 260 nm [33]. In this study, compound **1** was efficacious in inhibiting or killing the fungi by damaging their cell membranes, resulting in the leakage of the 260 nm and 280 nm absorbing materials, such as DNA, RNA, and proteins, which are essential for fungal growth.

To further investigate the mechanisms underlying compound **1**’s disruption of the cell membrane in *T. marneffei* cells, the Na^+^/K^+^-ATPase and Ca^2+^-ATPase activities of T. marneffei cells were detected. Na^+^/K^+^-ATPase is a carrier protein that exists in the phospholipid bilayer of cells. It mainly controls the transmembrane transport of Na^+^ and K^+^. It can release energy by decomposing ATP, and uses this energy to transport Na^+^ and K^+^ [34]. Ca^2+^-ATPase is a membrane transport protein ubiquitously found in the endoplasmic reticulum of all eukaryotic cells. As a calcium transporter, Ca^2+^-ATPase maintains a low cytosolic calcium level that enables a vast array of signaling pathways and physiological processes [35]. Na^+^/K^+^-ATPase and Ca^2+^-ATPase are important components of cell membrane transport. The Na+/K^+^-ATPase and Ca^2+^-ATPase activities of *T*. *marneffei* cells are shown in Figure 4E,F. Compared with the control group without **1**, the Na^+^/K^+^-ATPase and Ca^2+^-ATPase activities of the experimental group with compound **1** were significantly decreased. Among them, the experimental group with **1** at the levels of 1 MIC had the largest decrease. This showed that compound **1** had a certain inhibitory effect on the Na^+^/K^+^-ATPase and Ca^2+^-ATPase activities of *T. marneffei*, which is consistent with the above results.

These results indicated that the antibacterial action mode of compound **1** against *T. marneffei* probably involved the alteration of the structure of cell wall and membrane, causing the loss of cell viability.

### 2.5. Effect of Compound ***1*** on Morphology of T. marneffe

The morphological and ultrastructural changes in *T*. *marneffei* treated with compound **1** for 72 h were observed by SEM and TEM to better understand the antifungal mode of action of compound **1**. For *T*. *marneffei* cells, deformation was the most significant feature, which was apparent in the SEM image (Figure 5). The *T*. *marneffei* cells treated with compound **1** at 1/2 MIC displayed distorted membrane morphology, disruption of cell membrane, and leakage of cellular contents; and those treated with 1 MIC displayed distorted membrane morphology. Furthermore, a proportion of *T*. *marneffei* cells treated with compound **1** showed abnormalities in the TEM images, including the disappearance of cell wall, disruption of cell membrane, thinning of cytoplasm, distortion of cells, heterogeneous distribution of melanin, and leakage of intracellular materials (Figure 4G). S.K.P. Lau et al. reported that *T*. *marneffei* in yeast form can cause infections, and produce melanin as well, which plays an important role in the pathogenicity of *T*. *marneffei* [36]. Therefore, the decrease in intracellular melanin may also be the pathway of compound 1 inhibiting *T*. *marneffei*. These findings supported the results of the leakage of extracellular conductivity, nucleic acids and proteins leakage analysis, and Na^+^/K^+^-ATPase and Ca^2+^-ATPase activities in the present study.

### 2.6. Effects of Compound ***1*** on Mitochondrial Function

Mitochondria are key energy and metabolic regulatory centers within cells and also play an important role in maintaining cell growth and survival in mycelial cells. The core function of mitochondria is to synthesize ATP through oxidative phosphorylation. Therefore, the normal conduct of mitochondrial oxidative phosphorylation and the TCA cycle, especially the activities of related enzymes, is essential for maintaining cell survival. ATPase has an important role in energy metabolism [37]. The results of ATPase content in *T. marneffei* cells are shown in Figure 6A. The ATP content of the control group was 23,693.5 µmol/gprot. After treatment with 1 MIC compound **1**, the intracellular ATP level reduced to 8151.1 µmol/gprot, which was a 65.6% reduction (*p* < 0.05). In addition, after treatment with 1/2 MIC compound **1**, the intracellular ATP level reduced to 16,213.2 µmol/gprot, which was a 31.6% reduction (*p* < 0.05). ATPase content was significantly decreased after compound **1** treatment. These results indicated that the antifungal activity of compound **1** against *T. marneffei* can be attributed to disruption of the respiratory chain.

SDH (Succinate dehydrogenase) is a part of the respiratory chain (complex II). SDH catalyzes the oxidation of succinate to fumaric acid and FADH2. Therefore, it connects the TCA cycle with the respiratory chain, and the generated FADH2 does not dissociate from the enzyme, which directly uses the electrons to reduce the coenzyme Q, and then passes it to the complex III [38]. MDH (Malate dehydrogenase) can catalyze the reversible conversion between malic acid and oxaloacetate, and is also an important enzyme in mitochondrial function, which is mainly involved in some metabolic pathways such as photosynthesis, TCA cycle, and C4 cycle. Compared with the control, compound **1** resulted in a reduction in the activity of SDH and MDH. The SDH activity of compound **1** at the levels of 1/2 MIC and 1 MIC was significantly reduced compared with the control (*p* < 0.05) (Figure 6B), and the MDH activity of compound **1** at the levels of 1/2 MIC and 1 MIC was considerably lower than that of the control (*p* < 0.05) (Figure 6C). The above experimental results showed that MDH and SDH activities decreased with increasing compound **1** concentration in *T. marneffei* (Figure 6A,B), which suggested that compound **1** disrupts mitochondrial function by affecting MDH and SDH activities.

The above results suggested that compound **1** blocked the respiratory chain and energy metabolism, thereby killing the fungi.

## 3. Discussion

Polyene macrolide antibiotics are a significant group of antibiotics and have an important role in the treatment of fungal infections [39]. For example, amphotericin B, pimamycin, and nystatin have been widely used in clinical treatment [40]. Many researchers suggest that most polyene macrolides are bioactive compounds with a wide range of antifungal activity. However, the high hemolytic toxicity, poor water solubility, and unstable exposure to light limit the development of some compounds with good antifungal activity into clinical drugs. For a long time, researchers have been committed to chemical derivation, structural modification, genetic engineering, combined biosynthesis, and other methods to improve the antibacterial activity and solubility of these compounds and reduce the hemolytic toxicity. This has made certain research progress, but far from enough. Extensive research has shown that some bioactive secondary metabolites of marine microbial origin with strong antibacterial and antifungal activities are being intensely used as antibiotics and may be effective against infectious diseases [41]. In our study, four new compounds—tetrene macrolide compounds (**1**–**4**), and seven known polyene macrolide antibiotics, were isolated from the fermentation broth of the mangrove strain *S*. *hiroshimensis* GXIMD 06359. Their structures, including their relative configurations, were determined by HR-ESI-MS and NMR spectra. The antifungal activity of the compounds against *T. marneffei* was measured by detection of MIC and MBC in the present study. The results demonstrated that all compounds except compound **2** exhibit antifungal and fungicidal properties against *T*. *marneffei*. Moreover, the lowest MIC and MFC of the new compounds was compound **1**. To further analyze the antifungal activity of **1**, the time course of *T*. *marneffei* growth in the presence of **1** at different concentrations was plotted. These results demonstrated that **1** can significantly inhibit the growth of *T*. *marneffei*. Therefore, compound **1** could be considered an effective antibacterial agent; we further explore the mechanism of compound **1** inhibiting *T*. *marneffei*.

Cell membranes have important physiological functions, including maintaining the stability of the intracellular environment, signal transduction, and material transportation [42]. The integrity of cell membranes is crucial for cell viability, and membrane damage can lead to high cytotoxicity. Numerous studies have shown that the cell membrane of fungi is a target for inhibiting fungal growth and reproduction [43]. The previous studies have reported that the antifungal mechanism of action of polyene macrolides is binding to the fungal surface, which produces membrane breakdown, resulting in leakage of protein and vital nutrients and, ultimately, cell death [44,45,46]. In this study, it was found that compound **1** caused significant damage to cell membranes. Compound **1** irreversibly damaged the plasma membrane of *T*. *marneffei* cells. Its treatment increased extracellular conductivity, proteins, and nucleic acids in *T*. *marneffei* cultures, suggesting electrolyte leakage due to reduced membrane integrity of *T*. *marneffei* cells. Moreover, the decrease in Na^+^/K^+^-ATPase and Ca^2+^-ATPase activities confirmed the destruction of cell membrane function. SEM and TEM results confirmed that compound **1** treatment disrupted the integrity of *T. marneffei* cell walls and membranes. Cell wall and membrane integrity are critical for maintaining fungal viability. Kamble, M. T. et al. reported that SGF disrupted the bacterial cell membrane, resulting in leakage of intracellular biological components, and subsequently, cell death, in *Vibrio parahaemolyticus* and *Vibrio harveyi*, which is similar to our results [47].

In addition to destructing the cell membrane, mitochondrial dysfunction plays an important role in the potential mechanisms of antifungal drugs [48]. Xin et al. reported that antofine against *P. digitatum* is related to the cell membrane integrity and energy metabolism by affecting intracellular ATP content [49]. Pristimerin has been reported to exert antifungal activity; it caused mitochondrial membrane damage and affected mitochondria structure and functions, then oxidative phosphorylation and TCA cycle were inhibited, and energy metabolism was blocked in *S*. *sclerotiorum* [50]. In this study, we also investigated the role of mitochondrial function pathways against *T*. *marneffei.* Our results indicated that compound **1** caused a significant decrease in intracellular ATP levels and a significant decrease in the activities of MDH and SDH, and this was shown to be concentration-dependent. These results indicated that the antifungal activity of compound **1** against *T. marneffei* can be attributed to disruption of the respiratory chain. Therefore, the death of *T. marneffei* may be caused by mitochondrial dysfunction, in turn caused by the destruction of cytoplasmic membrane permeability and integrity.

This study showed that compound **1** effectively prevented *T. marneffei* growth. Compound **1** disrupts cytoplasmic membrane permeability and integrity, causes mitochondrial dysfunction, and *T. marneffei* metabolic disorders. We speculate that the antifungal mechanism of compound **1** on *T. marneffei* is through the destruction of *T. marneffei* cell membrane integrity and mitochondrial function to induce apoptosis. Compound **1** showed promising potential as a drug against *T. marneffei*. But the detailed role of compound **1** in bacterial membranes is unclear and needs further investigation.

## 4. Materials and Methods

### 4.1. General Experimental Procedures

TLC analyses were conducted on silica gel 60 F254-precoated plates. Silica gel 60 (200–300 mesh) were used for column chromatography (CC). For the HPLC analysis and purification, we used YMC C18 column (250 mm × 4.6 mm, 5 µm) and (250 mm × 10 mm, 5 µm). NMR spectra were recorded on Bruker AVANCE 500/125 spectrometer (Bruker, Fällanden, Switzerland) and Bruker AVANCE 700/175 spectrometer (Bruker, Hong Kong, China) with TMS as the internal standard.

### 4.2. Actinomycete Material

The strain *S. hiroshimensis* GXIMD 06359 was isolated from mangrove in the west coast of Hainan [24]. This strain is stored at Institute of Marine Drugs, Guangxi University of Chinese Medicine.

### 4.3. Fermentation, Extraction, and Isolation

After activation, *S*. *hiroshimensis* GXIMD 06359 was inoculated into 1 L flapper conical flask (containing 300 mL Am2ab medium, sterilized) and fermented in a constant temperature shaking table at 28 °C and 180 r/min for 10 days. After fermentation, the fermentation solution was filtered, and the bacterial solution and bacteria were separated. The bacterial solution was extracted with equal volume ethyl acetate three times, and the bacteria were soaked in equal volume acetone and extracted by ultrasound for 20 min until nearly colorless. The ethyl acetate phase and acetone phase fermentation crude extracts were obtained after concentration under reduced pressure. The crude extract (200.2 g) was subjected to normal phase silica gel column chromatography and gradient elution was performed with chloroform/acetone system (10:0, 10:2, 10:4, 10:8) and chloroform/methanol system (10:1, 10:2, 10:4, 0:10). The collected fractions were analyzed by thin layer chromatography (TLC) and HPLC. A total of 13 fractions (Fr. A1–A13) were obtained.

Fr. A10 (8.82 g) was separated by medium pressure preparative chromatography with ODS self-loaded column and gradient elution with methanol/water system at the flow rate of 15 mL/min. After the fraction was collected and analyzed by HPLC, a total of nine fractions (Fr. B1–Fr. B9) were obtained from the combined samples. Fr. B3 (0.2490 g) was separated by gel column chromatography, and three fractions (Fr. D1–D3) were obtained. Fr. D1 was separated and purified by semi-preparative HPLC. The mobile phase of Fr. D1 was methanol/water system [0–30 min: 42% methanol, 35–60 min: 55% methanol, compound **1** (3.6 mg, *t*_R_ = 36.85 min) and compound **6** (2.2 mg, *t*_R_ = 53.58 min) were obtained at the flow rate of 3 mL/min]. Compound **2** (1.5 mg, *t*_R_ = 63.45 min) was purified by semi-preparative HPLC from Fr. B4 (0.1772 g) by semi-preparative HPLC (48% methanol iso-degree elution for 40 min, flow rate 3 mL/min). Fr. B5 (2.7342 g) was isolated and purified by semi-preparative HPLC (0–30 min: 52% methanol, 35–60 min: 58% methanol, 65–100 min: 70% methanol, compound **5** (9.0 mg, *t*_R_ = 30.83 min), compound **3** (1.8 mg, *t*_R_ = 42.85 min), compound **4** (5.3 mg, *t*_R_ = 36.44 min), compound **7** (36.8 mg, *t*_R_ = 47.59 min), compound **8** (3.3 mg, *t*_R_ = 58.93 min), compound **9** (7.9 mg, *t*_R_ = 63.27 min), and compound **10** (1.6 mg, *t*_R_ = 71.65 min) were obtained at rate of 3 mL/min. Compound **11** (2 mg, *t*_R_ = 30.66 min) was purified by semi-preparative HPLC from Fr.B9 (82% methanol iso-degree elution for 40 min, flow rate 3 mL/min).

### 4.4. Microbial Strains OriginA and Culture Conditions

Reference strains (*Talaromyces marneffei* ATCC) were from YE Li from Guangxi Medical University (Guangxi Key Laboratory of AIDS Prevention and Control, School of Public Health, Guangxi Medical University). Seven-day-old pure culture of the yeast form grown on brain-heart infusion (BHI) agar was used in all reactions. The colonies of *T. marneffei* were flooded with Phosphate buffer saline (PBS) and the number of fungi was counted with a hemocytometer after washing three times. The cells were suspended in PBS and thoroughly vortexed. The suspensions were added to RPMI 1640 medium to obtain a stock of 1–5 × 10^6^ CFU/mL that was then diluted 1:100, resulting in a working stock of 1–5 × 10^4^ CFU/mL.

### 4.5. Antifungal Activity

Antifungal susceptibility testing was performed using the microdilution method according to CLSI protocol M27-A3 (Clinical and Laboratory Standards Institute) with minor modifications [51]. *Candida parpsilosis* ATCC 22019 was included as quality control through for all experiments. Wells containing inoculum alone and inoculum with DMSO were used as negative controls. AMB and FLC were used as a positive control. The minimum inhibitory concentration (MIC) was defined as the lowest concentration resulting in 100% inhibition of visible fungal growth after incubation at 37 °C for 72 h.

### 4.6. Determination of Minimal Fungicidal Concentration (MFC)

The MFCs of compounds were determined according to the methods of Mbah et al. [52]. Briefly, 10 µL from wells corresponding to 1, 2, 3, and 4-fold of the MIC, were placed on a Sabouraud Dextrose Agar (SDA) and incubated at optimal temperatures for 72 h. MFC was defined as the lowest concentration with no fungal growth.

### 4.7. Mode of Action of Compound ***1***

#### 4.7.1. Time-Kill Curve

Exponentially growing yeast cells were harvested and resuspended in RPMI-1640 to obtain a final concentration of 1–5 × 10^4^ CFU/mL. Different concentrations of compound **1** were added to the cells. Cells were incubated under shaking 180 rpm at 37 °C, and 10 µL from suspensions were placed on SDA and incubated at optimal temperatures for 72 h, then measured at the indicated time points after incubation (0, 12, 24, 36, 48, 60, and 72 h). The same volumes of solvents (DMSO) were added to the untreated controls. Three independent experiments were performed for optimal results.

#### 4.7.2. Scanning Electron Microscopy (SEM) and Transmission Electron Microscopy (TEM)

SEM was used to observe the morphological changes of compound **1**-treated *T. marneffei*. The fungal cells obtained from the logarithmic growth phase were treated with the compound **1** at 1/2 and 1-fold of the MIC value at 37 °C for 72 h. Then, the suspensions were centrifuged at 12,000 rpm/min for 10 min. The sediments were washed with 0.1 M PBS, (pH = 7.2) and fixed with 2.5% glutaraldehyde in PBS for 2 h at 4 °C. The cells were washed in the same buffer and were post-fixed for 30 min with osmium tetroxide. After harvesting, the cells were further dehydrated via graded ethanol concentrations (30%, 50%, 70%, 90%, and 100%) for 10 min each. Untreated cells were similarly processed and used as control. Then, cells were fixed on SEM support and observed by SEM (Sigma300, Zeiss), Wuhan, Hubei, China.

The pretreatment of fungal cells for transmission electron microscopy (TEM) were the same as that for scanning electron microscopy (SEM, Wuhan, Hubei, China). After being fixed with 2.5% glutaraldehyde, post-fixed by 1% osmic acid, dehydrated using alcohol, permeated using white resin, and embedded by roasting at 55 °C, the samples were cut into thin sections to perform TEM (HITACHI HT 7800 120 kv, Wuhan, Hubei, China).

#### 4.7.3. Leakage of Extracellular Conductivity

Fungal membrane permeability was determined and expressed as the electric conductivity according to the method by Maliehe, T. S. et al. [53]. Fungal cells were cultivated at 37 °C to mid-exponential stage and collected by centrifugation (8000 rpm for 15 min). Cells were washed twice in 0.1 M PBS. The different concentrations compound **1** were added into the isotonic fungal suspensions (1 × 10^4^ CFU/mL) and incubated at 37 °C for 15 h. Thereafter, their conductivities were measured and recorded as A_1_ (0, 3, 6, 9, 12, 15 h). The conductivities of the fungi in 0.1 M PBS treated with boiling water for 5 min were used as the control and marked as A_0_. The cell membrane permeability was then calculated using the formula: Electric conductivity = A_1_ − A_0_.

#### 4.7.4. Leakage of 260 nm and 280 nm Absorbing Material

Fungal strains were cultured in RPMI-1640 and incubated at 37 °C for 12 h. The most active compound **1** were added to the fungal suspensions at 1-fold and 1/2-fold of the MIC values. Suspensions were incubated at 37 °C and samples were removed at times 0, 2, 4, 6, 8, and 10 h and centrifuged at 10,000× *g* for 10 min at 4 °C. 200 µL of supernatants from each condition were added to a 96-well plate. Wells and absorbance values at 260 nm and 280 nm were recorded using a UV spectrophotometer. The following controls were included: a fungal suspension in RPMI-1640 without antimicrobial agents as the negative control; a fungal suspension with AMB as the positive controls.

#### 4.7.5. Detection of Na^+/^K^+^-ATPase and Ca^2+^-ATPase

The fungal cells obtained from the logarithmic growth phase were treated with the compound **1** at ½ and 1-fold of the MIC value at 37 °C for 72 h. Drug-treated fungal solutions were rinsed with sterile PBS and resuspended (1 × 10^4^ cells mL^−1^). The activities of Na^+/^K^+^-ATPase and Ca^2+^ -ATPase were analyzed using commercial kits (NanJing JianCheng, Nanjing, China) according to the instructions.

#### 4.7.6. Measurement of Intracellular ATPase Concentration

The fungal cells obtained from the logarithmic growth phase were treated with the compound **1** at ½ and 1-fold of the MIC value at 37 °C for 72 h. Drug-treated fungal solutions were rinsed with sterile PBS and resuspended (1 × 10^4^ cells mL^−1^). The intracellular ATPase concentration was determined using the ATP assay kit (NanJing JianCheng, Nanjing, China).

#### 4.7.7. Detection of MDH and SDH

The fungal cells obtained from the logarithmic growth phase were treated with the compound **1** at ½ and 1-fold of the MIC value at 37 °C for 72 h. Drug-treated fungal solutions were rinsed with sterile PBS and resuspended (1 × 10^4^ cells mL^−1^). The activities of MDH and SDH were analyzed using commercial kits (NanJing JianCheng, Nanjing, China) according to the instructions.

## Figures and Tables

**Figure 1 marinedrugs-22-00038-f001:**
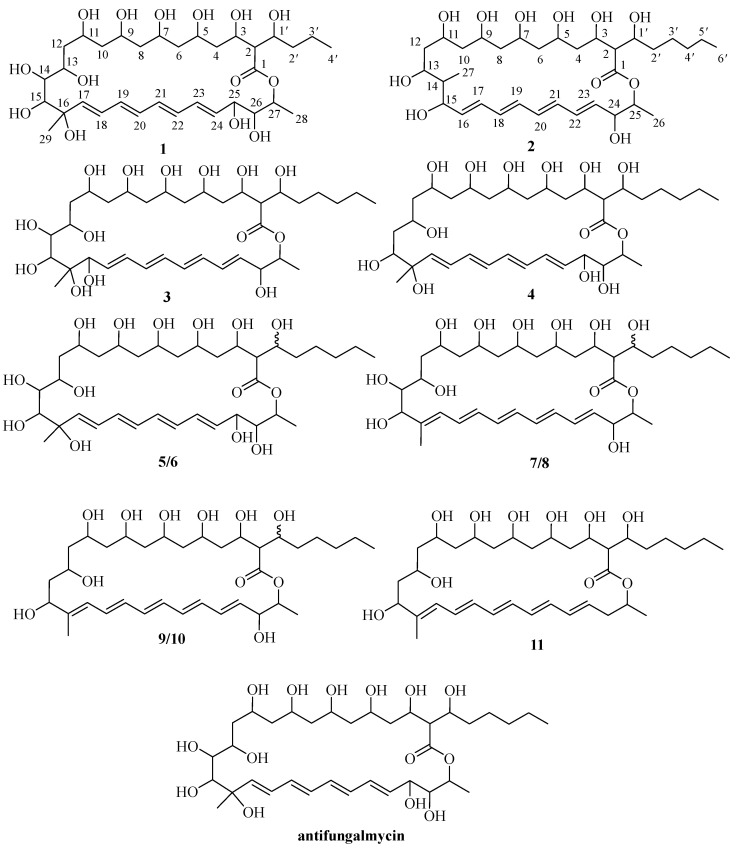
Chemical structures of compounds **1**–**11** and antifungalmycin.

**Figure 2 marinedrugs-22-00038-f002:**
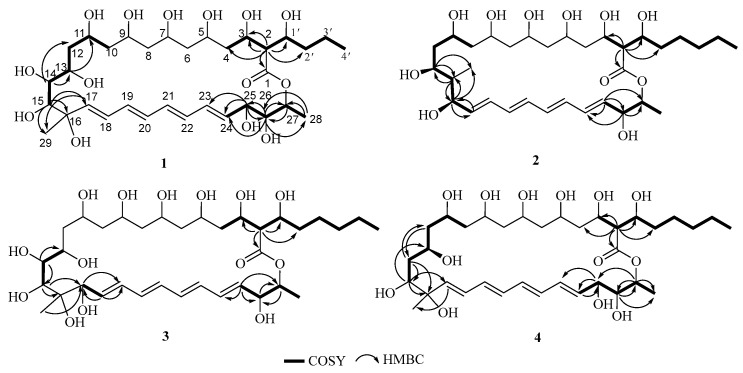
The key ^1^H-^1^H COSY, HMBC correlations of compounds **1**–**4**.

**Figure 3 marinedrugs-22-00038-f003:**
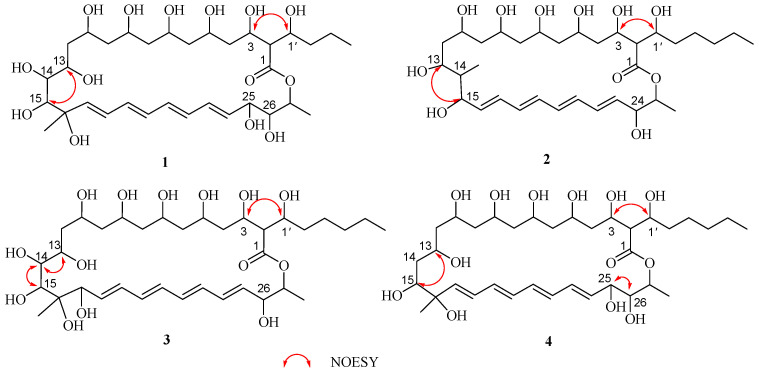
The key NOESY correlations of compounds **1**–**4.**

**Figure 4 marinedrugs-22-00038-f004:**
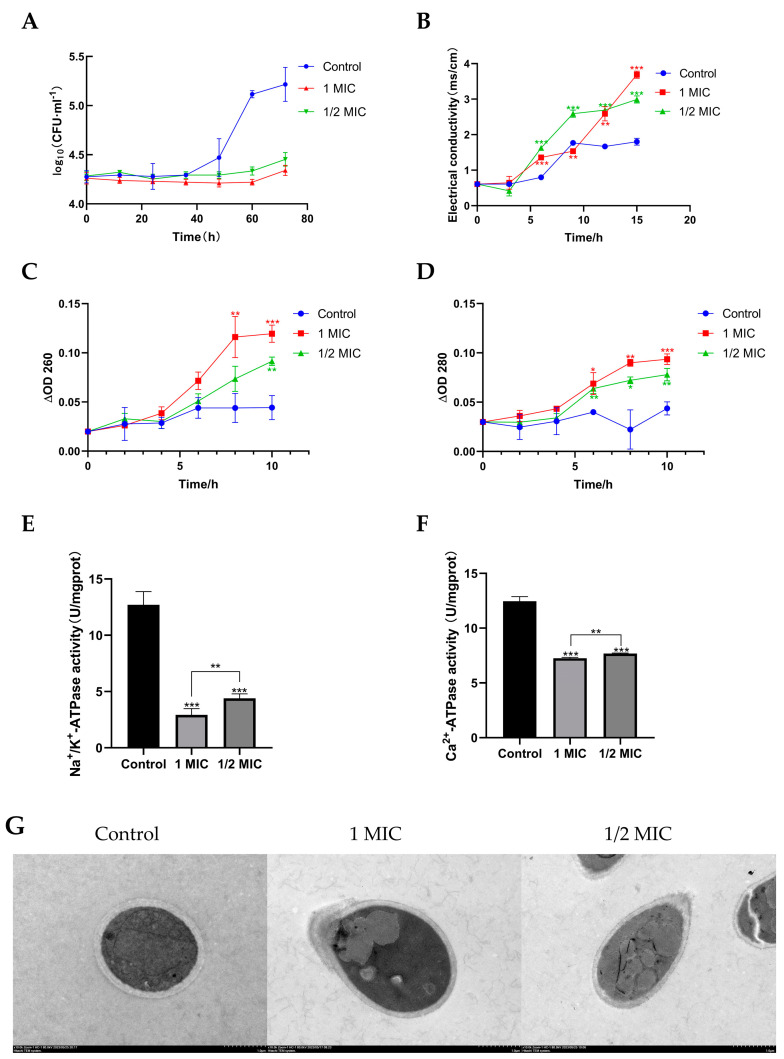
(**A**) Time-kill curve of compound **1** against *T. marneffei*. (**B**) The effect of compound **1** on the extracellular conductivity of *T. marneffei*. (**C**) The effect of compound **1** on the extracellular nucleic acids of *T. marneffei*. (**D**) The effect of compound **1** on the extracellular proteins of *T. marneffei*. (**E**) The effect of compound **1** on Na^+/^K^+^-ATPase activity against *T. marneffei*. (**F**) The effect of compound **1** on Ca^2+^-ATPase activity against *T. marneffei*. (**G**) TEM of *T. marneffei* treated with compound **1** for 72 h. Data are shown as mean ± S.D. * *p* < 0.05, ** *p* < 0.01, and *** *p* < 0.001; Student’s *t* test, *n* = 3.

**Figure 5 marinedrugs-22-00038-f005:**
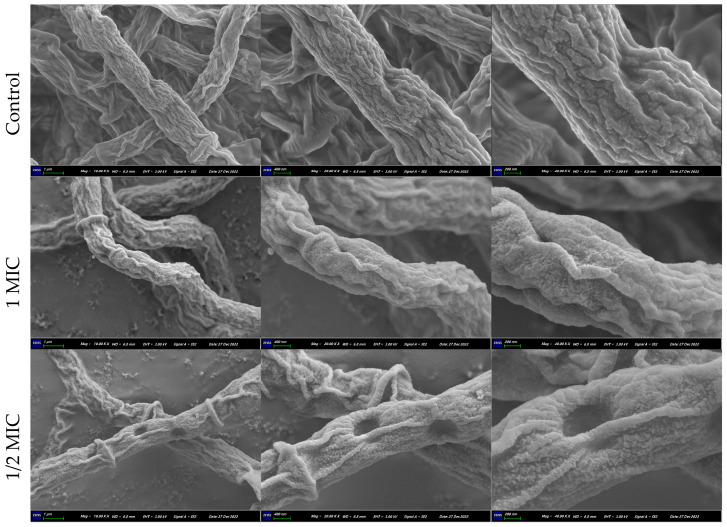
SEM of *T. marneffei* treated with compound **1** for 72 h.

**Figure 6 marinedrugs-22-00038-f006:**
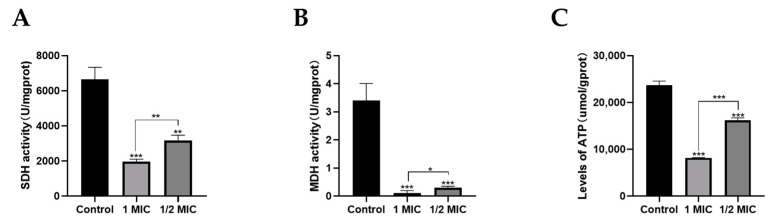
(**A**) The effect of compound **1** on ATPase content against *T. marneffei*. (**B**) The effect of compound **1** on SDH activity against *T. marneffei*. (**C**) The effect of compound **1** on MDH activity against *T. marneffei*. Data are shown as mean ± S.D. * *p* < 0.05, ** *p* < 0.01, and *** *p* < 0.001; Student’s *t* test, *n* = 3.

**Table 1 marinedrugs-22-00038-t001:** ^1^H NMR and ^13^C NMR data for compounds **1-4** in DMSO-*d_6_*.

NO.	1 ^a^		2 ^a^		3 ^b^		4 ^b^	
	*δ*_H_ (*J* in Hz)	*δ* _C_	*δ*_H_ (*J* in Hz)	*δ* _C_	*δ*_H_ (*J* in Hz)	*δ* _C_	*δ*_H_ (*J* in Hz)	*δ* _C_
1	-	171.6, C	-	171.2, C	-	171.3, C	-	171.7, C
2	2.46, dd (8.6, 6.8)	58.3, CH	2.38, t (8.0)	58.5, CH	2.48, m	58.2, CH	2.46, dd (8.8, 6.6)	58.2, CH
3	4.02, m	69,6, CH	4.05, m	69,7, CH	4.05, m	70.0, CH	4.03, m	69,4, CH
4	1.52, m	40.9, CH_2_	1.38, m	40.9, CH_2_	1.45, m	43.3, CH_2_	1.50, m	40.9, CH_2_
5	3.90, m	67.8, CH	3.93, m	70.2, CH	3.93, m	68.2, CH	3.92, m	69.4, CH
6	1.54, m	44.3, CH_2_	1.40, m	44.0, CH_2_	1.45, m	45.4, CH_2_	1.50, m	44.1, CH
7	3.90, m	68.5, CH	3.93, m	70.3, CH	3.98, m	69.0, CH	3.92, m	68.2, CH
8	1.47, m	44.5, CH_2_	1.50, m	43.7, CH_2_	1.46, m	46.3, CH_2_	1.50, m	44.6, CH_2_
9	3.97, m	69.2, CH	3.93, m	69.8, CH	3.93, m	67.0, CH	3.97, m	68.2, CH
10	1.71, m; 1.25, m	42.2, CH_2_	1.70, m; 1.25, m	42.9, CH_2_	1.76, m;1.45, m	43.6, CH_2_	1.76, m; 1.53, m	41.4, CH_2_
11	3.90, m	67.8, CH	3.93, m	66.7, CH	3.93, m	66.9, CH	3.87, m	67.5, CH
12	1.62, m	41.6, CH_2_	1.92, m; 1.52, m	38.5, CH_2_	3.33, m	63.1, CH_2_	1.74, m; 1.50, m	44.4, CH_2_
13	3.29, m	75.2, CH	3.21, m	80.5, CH	3.46, m	66.0, CH	3.65, m	71.2, CH
14	3.34, m	82.3, CH	2.87, m	73.0, CH	3.80, m	84.6, CH	2.16, m	41.9, CH_2_
15	3.65, d (5.4)	82.7, CH	3.63, m	80.5, CH	3.71, d	78.6, CH	3.90, m	76.0, CH
16	-	82.0, C	5.89, dd (14.0, 3.1)	129.5, CH	-	80.0, C		84.5, CH
17	5.72, d (15.0)	138.9, CH	6.20, m	133.3, CH	4.18, d (4.4)	83.4, CH	5.65, d (15.0)	137.8, C
18	6.15, dd (15.1, 9.4)	126.3, CH	6.18, m	130.4, CH	5.72, dd (14.5, 4.3)	129.1, CH	6.15, dd (14.5, 6.2)	127.4, CH
19	6.24, m	131.5, CH	6.28, m	131.4, CH	6.31, m	131.1, CH	6.22, m	130.8, CH
20	6.26, m	133.2, CH	6.26, m	130.7, CH	6.26, m	130.4, CH	6.21, m	131.1, CH
21	6.28, m	131.2, CH	6.23, m	131.1, CH	6.31, m	131.5, CH	6.21, m	133.5, CH
22	6.28, m	133.5, CH	6.20, m	133.8, CH	6.26, m	131.2, CH	6.32, m	134.0, CH
23	6.28, m	130.4, CH	5.79, dd (14.7, 6.8)	135.9, CH	6.23, m	133.1, CH	6.25, m	130.3, CH
24	5.67, dd (14.6, 5.8)	134.1, CH	3.82, m	73.3, CH	6.36, m	133.9, CH	5.68, dd (14.6, 6.1)	134.4, CH
25	4.24, dd (5.7, 2.3),	72.0, CH	4.62, m	72.0, CH	5.99, dd (15.1,5.1)	136.1, CH	4.23, m	72.1, CH
26	3.51, dd (7.2, 2.6)	75.7, CH	1.19, d (6.3)	17.8, CH_3_	3.87, m	72.4, CH	3.50, m	75.5, CH
27	4.71, m	70.3, CH	0.84, m	15.6, CH_3_	4.57, dd (8.9, 6.3)	72.5, CH	4.77, m	70.2, CH
28	1.14, d (6.2)	17.4, CH_3_	-	-	1.21, d (6.2)	17.9, CH_3_	1.14, d (6.3)	17.5, CH_3_
29	1.06, s (7.1)	22.9, CH_3_	-	-	1.20, s	19.6, CH_3_	1.10, s	22.2, CH_3_
1′	3.68, m	69.2, CH	3.72, m	69.6, CH	3.68, m	69.7, CH	3.65, m	69.2, CH
2′	1.29, m; 1.24, m	36.7, CH_2_	1.34, m; 1.23, m	33.9, CH_2_	1.45, m; 1.25, m	34.3, CH_2_	1.33, m; 1.24, m	34.5, CH_2_
3′	1.44, m; 1.25, m	18.2, CH_2_	1.42, m; 1.24, m	24.4, CH_2_	1.45, m; 1.25, m	24.6, CH_2_	1.42, m; 1.24, m	24.5, CH_2_
4′	0.83, t (7.0)	14.0, CH_3_	1.24, m	31.3, CH_2_	1.24, m	31.3, CH_2_	1.24, m	31.3, CH_2_
5′	-	-	1.24, m	22.1, CH_2_	1.25, m	22.1, CH_2_	1.25, m	22.1, CH_2_
6′	-	-	0.84, t (6.9)	14.0, CH_3_	0.85, t (6.9)	14.0, CH_3_	0.85, t (7.1)	14.0, CH_3_
3-OH	5.12, s	-	5.06, m	-	5.23, d	-	5.10, d	-
5-OH	4.85, m	-	4.90, d	-	4.94, d	-	4.88, m	-
7-OH	4.85, m	-	5.04, m	-	4.82, d	-	4.88, m	-
9-OH	4.90, m	-	5.07, m	-	4.32, d	-	4.86, m	-
11-OH	4.85, m	-	4.56, d	-	4.40, d	-	4.68, d	-
13-OH	-	-	4.78, d	-	4.71, m	-	4.80, m	-
14-OH	-	-	-	-	4.65, d	-	-	-
15-OH	-	-	4.78, m	-	4.92, d	-	4.88, m	-
16-OH	-	-	-	-	4.48, d	-	-	-
17-OH	-	-	-	-	5.00, m	-	-	-
24-OH	-	-	5.26, d	-	-	-	-	-
25-OH	-	-	-	-	-	-	4.86, m	-
26-OH	-	-	-	-	5.29, d	-	4.91, d	-
1′-OH	4.85, m	-	4.82, d	-	4.85, d	-	4.80, m	-

^a^ The ^1^H NMR measured at 500 MHz and ^13^C NMR measured at 175 MHz. ^b^ The ^1^H NMR measured at 700 MHz and ^13^C NMR measured at 175 MHz.

**Table 2 marinedrugs-22-00038-t002:** MIC and MFC of the compounds.

Compounds	MIC (μg/mL)	MFC (μg/mL)
**1**	16	64
**2**	>128	——
**3**	128	——
**4**	32	——
**5**	32	64
**6**	128	——
**7**	4	8
**8**	16	16
**9**	2	4
**10**	32	128
**11**	32	——
FLC *	16	64
AMB *	0.5	1

* FLC and AMB serve as positive controls. FLC, Fluconazole; AMB, Amphotericin B. “——" means compounds is not enough to support the experiment, no detection.

## Data Availability

The data presented in this study are available upon request from the corresponding author.

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
