# Peer review of "New Polyene Macrolide Compounds from Mangrove-Derived Strain Streptomyces hiroshimensis GXIMD 06359: Isolation, Antifungal Activity, and Mechanism against Talaromyces marneffei"

_marinedrugs, 2024, doi:10.3390/md22010038_

Round 1
Reviewer 1 Report
Comments and Suggestions for Authors
In this manuscript, four new polyene macrolides and seven known analogues isolated from a marine-derived Streptomyces strain were reported. The structural elucidation of these compounds is extremely challenging, and their antifungal activity is also very attractive. Thus, I suggest accepting this manuscript for publication. However, before that, some issues must be solved.
1. The title can be changed to "New Polyene Macrolides from the Mangrove-Derived Streptomyces hiroshimensis GXIMD 06359:Isolation, Antifungal Activity and Mechanism Against Talaromyces marneffei".
2. The asterisk in Figure 1 is not necessary.
3. The NOESY correlations mentioned in lines 97, 130, 165, and 198 can not be used as evidence for determining relative configurations. The relative configurations around those structural units need more precise evidence. So, I suggest moving the part of the relative configuration determination. The corresponding figures should also be revised.
Author Response
Response to Reviewer 1 Comments
Thank you very much for taking the time to review this manuscript. Please find the detailed responses below and the corresponding revisions/corrections highlighted/in track changes in the re-submitted files.
Point 1: The title can be changed to "New Polyene Macrolides from the Mangrove-Derived Streptomyces hiroshimensis GXIMD 06359:Isolation, Antifungal Activity and Mechanism Against Talaromyces marneffei".
Response 1: Thank you for your suggestion. We have changed the title to “New Polyene Macrolides from the Mangrove-Derived Streptomyces hiroshimensis GXIMD 06359:Isolation, Antifungal Activity and Mechanism Against Talaromyces marneffei”.(lines 2-5)
Point 2: The asterisk in Figure 1 is not necessary.
Response 2: Thank you for your suggestion. The asterisk in Figure 1 have been deleted. (Figure 1)
Point 3: The NOESY correlations mentioned in lines 97, 130, 165, and 198 can not be used as evidence for determining relative configurations. The relative configurations around those structural units need more precise evidence. So, I suggest moving the part of the relative configuration determination. The corresponding figures should also be revised.
Response 3: Thank you for your suggestion. As you commented, the NOESY correlations do not infer the relative configuration of the compound, we have removed the relevant representation, and made a new description of this part of the content and made corresponding corrections in the figures. (lines 96-99, 129-131, 161-163, 192-195)
Reviewer 2 Report
Comments and Suggestions for Authors
The manuscript entitled “New Polyene Macrolides Compounds from Mangrove-Derived Strain Streptomyces hiroshimensis GXIMD 06359:Isolating, Antifungal Activity and Mechanism Against Talaromyces marneffei” reports the isolation of four new macrolide polyene antibiotics, the antifungalalmycin B-E and seven known analogues from the fermentation broth of the mangrove strain Streptomyces hiroshimensis GXIMD 06359. Molecular structures of the isolated compounds were confirmed by NMR, UV, and HR-ESI-MS. Moreover, the title compounds were also evaluated in vitro for their antifungal activity against Talaromyces marneffei.
In general, the manuscript presents several new results, and merit to be published in marinedrugs. But it needs some revisions, according to the following comments:
1. Some corrections should be made (lack of space, forgotten points to add or others to delete)? to check. Example: Lines 245, 248, 480, …
2. Line 102, “m/z” must be in italics.
3. Line 208, “antifungalmycin a” should be “antifungalmycin A”
4. Lines 209 and 216, “antifungalmycin b” should be “antifungalmycin B”
5. Line 231, “Compound 9” should be “compound 9”
6. Line 232, “Compound 1” should be “compound 1”
7. In table 5, abbreviations “FLC” and “AMB” must be specified.
8. Lines 349 and 353, abbreviations “SDH” and “MDH” should be specified.
9. Besides, authors should provide HRMS images (in Supplementary data) for purity verification.
Author Response
Response to Reviewer 2 Comments
Thank you very much for taking the time to review this manuscript. Please find the detailed responses below and the corresponding revisions/corrections highlighted/in track changes in the re-submitted files.
Point 1: Some corrections should be made (lack of space, forgotten points to add or others to delete)? to check. Example: Lines 245, 248, 480, …
Response 1: Thank you for your suggestion. We have carefully checked the manuscript, and all of the errors have been corrected in text.
Point 2: Line 102, “m/z” must be in italics.
Response 2: Thank you for your suggestion. We have changed “m/z” in italics. (line 101)
Point 3: Line 208, “antifungalmycin a” should be “antifungalmycin A”.
Response 3: Thank you for your suggestion. Because of the NMR data of compounds 5 and 6 were similar to the previously reported antifungalmycin in the literature. The above difference in chemical shift may be caused by the different stereoscopic configuration of C-3 and C-1'. However, due to the small amount of compound 6, which was unstable and changeable, we cannot determine the relative configuration of C-1'. Based on this, we believed that the planar structures of compounds 5 and6 were consistent with the reported antifungalmycin. If other members of our research group can isolate more amount of these compounds in the follow-up study, we will determine the chiral center again. Therefore, compounds 5 and 6 were renamed antifungalmycin a1 and antifungalmycin a2 to avoid confusion. (lines 207-208)
Point 4: Lines 209 and 216, “antifungalmycin b” should be “antifungalmycin B”.
Response 4: Thank you for your suggestion. The NMR data of compounds 5 and 6 were similar to the previously reported antifungalmycin. The main differences between of 5 and 6 in chemical shift may be caused by the different stereoscopic configuration of C-3 and C-1'. However, due to the small amount of compound 6, which was unstable and changeable, we cannot determine the relative configuration of C-1'. Based on this, we believed that the planar structure of compounds 5 and 6 were consistent with the reported antifungalmycin. Therefore, compounds 5 and 6 were renamed antifungalmycin a1 and antifungalmycin a2 to avoid confusion. The same response applies to compounds 7 and 8, 9 and 10. (lines 207-208, 215, 221-222)
Point 5: Line 231, “Compound 9” should be “compound 9”.
Response 5: Thank you for your suggestion. We have changed “Compound 9” to “compound 9”. (line 230)
Point 6: Line 232, “Compound 1” should be “compound 1”.
Response 6: Thank you for your suggestion. We have changed “Compound 1” to “compound 1”. (line 231)
Point 7: In table 5, abbreviations “FLC” and “AMB” must be specified.
Response 7: Thank you for your suggestion. We have specified “FLC” and “AMB” in table 2.
Point 8: Lines 349 and 353, abbreviations “SDH” and “MDH” should be specified.
Response 8: Thank you for your suggestion. We have specified “SDH” and “MDH”. (lines 346, 350)
Point 9: Besides, authors should provide HRMS images (in Supplementary data) for purity verification.
Response 9: Thank you for your suggestion. We have provided HRMS images in Supplementary data.
Reviewer 3 Report
Comments and Suggestions for Authors
The manuscript describes the isolation, structure elucidation and antifungal activity of polyene macrolides from Streptomyces hiroshimensis. The results, especially the isolation of four new macrolides from S. hiroshimensis and the antifungal activity of antifungalmycin B, are excellent. However, the structure elucidation part is weak and requires further discussion on how the author came to define the stereochemistry of various chiral centers.
Major comments –
· The observed NOESY correlations between H-3/H-1’, H-13/H15, H-25/H-26 (lines 96-99) don’t allow to define the relative stereochemistry of remaining chiral centers i.e., C-%, C-7, C-9, C-11. No explanation was given as to why chiral center C-14 remain undefine for antifungalmycin B and C (1 and 2). It is worth mentioning here that reference 25 (Natural Products and Bioprospecting, 2012, 2, 41-45) cited in the manuscript only described the planner structure of antifungalmycin and reference 26 [PLoS One 2013, 8(8), e73884] have simply described stereostructure of antifungalmycin 702 without structure elucidation.
· The abstract indicated that macrolides 5-11 were known compounds, yet the author provided names for these macrolides (lines 208-209, 215-216, 222-223) without any references. The paragraph (lines 202-207) specified that compounds 5 and 6 possess different stereochemistry than antifungalmycin and antifungalmycin 702, as reported in references 26 and 26. This suggests that both compounds are likely to be new natural products. The same principle applies to compounds 7, 8, 9, and 10.
· The names of the new macrolide 1 (antifungalmycin B) and compound 7 (antifungalmycin b) are nearly identical, differing only in the capitalization of a single letter. It would be preferable to assign distinct names using different letters to avoid confusion.
Minor comments -
· Although the author closely compared the proton and carbon NMR spectral data of macrolides 1-4 with antifungalmycin to determine the lactone ring structure, the structure of antifungalmycin was not shown. It is encouraged the author to incorporate the structure of antifungalmycin or antifungalmycin 702 as described in the references in Figure 1.
· In Figure 1, macrolides 1-4 are marked with asterisks without any description in the legend.
· Antifungalmycin B should have the hydroxy group at C-14 instead of the methyl group (Figure 1-3).
· It is better to present proton and carbon NMR data of all macrolides in one Table that makes it easy to compare.
Author Response
Response to Reviewer 3 Comments
Thank you very much for taking the time to review this manuscript. Please find the detailed responses below and the corresponding revisions/corrections highlighted/in track changes in the re-submitted files.
Point 1: The observed NOESY correlations between H-3/H-1’, H-13/H15, H-25/H-26 (lines 96-99) don’t allow to define the relative stereochemistry of remaining chiral centers i.e., C-%, C-7, C-9, C-11. No explanation was given as to why chiral center C-14 remain undefine for antifungalmycin B and C (1 and 2). It is worth mentioning here that reference 25 (Natural Products and Bioprospecting, 2012, 2, 41-45) cited in the manuscript only described the planner structure of antifungalmycin and reference 26 [PLoS One 2013, 8(8), e73884] have simply described stereostructure of antifungalmycin 702 without structure elucidation.
Response 1: Thank you for your comment. All of the compounds, the chiral centers C-5, C-7, C-9 and C-11 have very close chemical shifts of protons, or even overlap. Therefore, it was very difficult to observe their NOESY signals and determine their relative configuration. Thus, the relative configuration of this part cannot be determined. In compound 1, we do not observe the NOESY correlations signal of H-14 in its NOESY spectra, so its relative configuration cannot be determined. We have removed the part of the relative configuration determination and rephrased the section. (lines 96-99, 129-131, 161-163, 192-195)
Point 2: The abstract indicated that macrolides 5-11 were known compounds, yet the author provided names for these macrolides (lines 208-209, 215-216, 222-223) without any references. The paragraph (lines 202-207) specified that compounds 5 and 6 possess different stereochemistry than antifungalmycin and antifungalmycin 702, as reported in references 26 and 26. This suggests that both compounds are likely to be new natural products. The same principle applies to compounds 7, 8, 9, and 10.
Response 2: Thank you for your comment. During the experiment, relatively large amounts of compounds 5, 7, and 9 were isolated. The NMR data of compound 5, 7 and 9 were greatly similar to the reported compounds antifungalmycin, fungichromin and filipin III, respectively. At the same time, we also isolated small amount of compounds 6, 8, and 10, because of the amount of these compounds were too small, and these compounds were unstable and changeable, there was no way to determine the relative configuration of the C-1' position of these compounds by the existing amount. Thus, these compounds were not written as new compounds. At the same time, because the research content of this part was relatively significant, other members of our research group are still conducting relevant research. We expect to get more quantity in the follow-up studies to determine the relative configuration of these compounds. (lines 207-208, 215, 161-163, 221-222)
Point 3: The names of the new macrolide 1 (antifungalmycin B) and compound 7 (antifungalmycin b) are nearly identical, differing only in the capitalization of a single letter. It would be preferable to assign distinct names using different letters to avoid confusion.
Response 3: Thank you for your suggestion. We have renamed the compounds 5, 6, 7, 8, 9 and 10 to avoid confusion. (lines 207-208, 215, 161-163, 221-222)
Point 4: Although the author closely compared the proton and carbon NMR spectral data of macrolides 1-4 with antifungalmycin to determine the lactone ring structure, the structure of antifungalmycin was not shown. It is encouraged the author to incorporate the structure of antifungalmycin or antifungalmycin 702 as described in the references in Figure 1.
Response 4: Thank you for your suggestion. We have shown the structure of antifungalmycin in Figure 1。
Point 5: In Figure 1, macrolides 1-4 are marked with asterisks without any description in the legend.
Response 5: Thank you for your comment. The asterisks in Figure 1 have been deleted.
Point 6: Antifungalmycin B should have the hydroxy group at C-14 instead of the methyl group (Figure 1-3).
Response 6: Thank you for your comment. After careful verification, Antifungalmycin B realy has the hydroxy group at C-14. Antifungalmycin B (compound 2) has a methyl group rather than a hydroxyl group at the C-14.
Figure 1. HMBC spectrum of compound 2.
Point 7: It is better to present proton and carbon NMR data of all macrolides in one Table that makes it easy to compare.
Response 7: Thank you for your suggestion. We have presented proton and carbon NMR data of all macrolides in Table 1.

Round 2
Reviewer 3 Report
Comments and Suggestions for Authors
The revised manuscript addresses all the queries. I would like to recommend it for publication in Marine Drugs after minor revision.
Compound 2 is a 26-member macrolide; it would be beneficial to include numbering on the structure in Figure 1.
Verify the chemical shift of C-14 of compound 2 is 73.0 ppm (Table 1).
Author Response
Response to Reviewer 3 Comments (Round 2)
Thank you very much for taking the time to review this manuscript. Please find the detailed responses below and the corresponding revisions/corrections highlighted/in track changes in the re-submitted files.
Point 1: Compound 2 is a 26-member macrolide; it would be beneficial to include numbering on the structure in Figure 1.
Response 1:Thank you for your suggestion. We have edited the numbering on the structure in Figure 1.
Point 2: Verify the chemical shift of C-14 of compound 2 is 73.0 ppm (Table 1).
Response 2: Thank you for your suggestion. After careful verification, the chemical shift of C-14 of compound 2 is 73.0 ppm. (Figure 1)